# Depression and cancer outcomes in resource-limited settings: A cross-sectional analysis of treatment uptake and survival among metastatic breast cancer patients in Lagos, Nigeria

Emma Wisniewski[1], Adewumi Alabi[2,3], Abdulrazzaq Lawal[4], Anthonia Sowunmi[2,3], Bolanle Adegboyega[2,3], Chen-Chih Chung[5,6], Sumaiya Nezam[7], Sylvia Shirima[8], Donaldson F. Conserve[7], Oluwaseun Adebayo Bamodu[8,9]*

1 Department of Global Health, Milken Institute School of Public Health, George Washington University, Washington, District of Columbia, United States of America, 2 Department of Radiation Biology, Radiotherapy & Radiodiagnosis, College of Medicine, University of Lagos, Lagos, Nigeria, 3 NSIA-LUTH Cancer Centre, Lagos University Teaching Hospital, Idi-Araba, Lagos, Nigeria, 4 Department of Surgery, College of Medicine, University of Lagos, Lagos, Nigeria, 5 Department of Neurology, Taipei Medical University, Shuang Ho Hospital, New Taipei City, Taiwan, 6 Department of Neurology, School of Medicine, College of Medicine, Taipei Medical University, Taipei City, Taiwan, 7 Department of Community Health and Prevention, Milken Institute School of Public Health, George Washington University, Washington, District of Columbia, United States of America, 8 Building Research and Implementation to Drive Growth and Equity in Africa (BRIDGEAfrica), Dar Es Salaam, Tanzania, 9 Bethesda Achievers Consultancy, Hagerstown, Maryland, United States of America

* dr_bamodu@yahoo.com

## Abstract

The intersection of depression and cancer outcomes represents a critical yet under-studied domain in global oncology, particularly in low- and middle-income countries (LMICs) where both conditions impose substantial disease burdens. While evidence from high-income countries demonstrates associations between depression and adverse cancer outcomes, the generalizability of these findings to resource-limited settings remains uncertain. Theoretical frameworks including Beck's cognitive model and the biopsychosocial model suggest multiple pathways through which depression may influence cancer outcomes; however, these pathways may operate differently in resource-constrained healthcare environments. This study examines the relationship between clinically significant depression (CSD) and treatment uptake and survival among metastatic breast cancer patients in Nigeria, providing novel insights into psycho-oncology within the sub-Saharan African context. We conducted a cross-sectional analysis of 313 metastatic breast cancer patients presenting to NSIA-LUTH Cancer Centre, Lagos, Nigeria (September 2020–February 2022). Depression was assessed using the Beck Depression Inventory-II (BDI-II), with CSD defined as BDI-II ≥ 20, consistent with oncology-validated cutoffs and African psychometric valida-tion studies. Associations between CSD and treatment receipt were evaluated using

**Data availability statement:** The De-identified participant data may be made available upon reasonable request and approval by the institutional ethics committees, subject to data sharing agreements that ensure participant privacy and appropriate use of data. The de-identified dataset cannot be publicly deposited due to ethical and legal restrictions. Our Institutional Review Board (Lagos University Teaching Hospital Health Research Ethics Committee, IRB ADM/DCST/HREC/APP/7058) explicitly prohibits unrestricted public data sharing for this vulnerable population (women with terminal metastatic breast cancer), as this would violate Nigerian data protection regulations and could compromise patient privacy through indirect identification despite de-identification. Public deposition would violate our IRB-approved protocol and participant consent, which specified controlled data sharing only as imposed by the Lagos University Teaching Hospital Health Research Ethics Committee (LUTHHREC). However, the complete de-identified dataset is available to qualified researchers upon reasonable request through a controlled access mechanism. This study represents a secondary analysis of de-identified data from the primary study "Access to psychological support, psycho-oncology and psychotherapy among cancer patients in Nigeria" (Principal Investigator: Dr. A.O. Alabi). Data are available from the LUTH-HREC Data Access Committee for researchers who meet criteria for access to confidential data. Institutional Contact for Data Access: Lagos University Teaching Hospital Health Research Ethics Committee (LUTHHREC) Idi-Araba, Lagos, Nigeria Email: cmd@lasuth.org; info@luth.gov.ng Phone: +234 812 836 4824 IRB Approval Number: ADM/DCST/HREC/APP/7058 Data access requests will be reviewed by the LUTH-HREC to ensure compliance with Nigerian research ethics regulations and patient privacy protections. Qualified researchers must submit: (1) institutional affiliation and qualifications, (2) intended use and analysis plan, and (3) formal data sharing agreement. Approved requests typically processed within 2-4 weeks, with data transfer within 6 months. This controlled access approach complies with Nigerian research ethics regulations, protects

multivariable logistic regression. Survival patterns were examined using Kaplan-Meier analysis and Cox proportional hazards models, adjusting for clinicopathological confounders. The cross-sectional design precludes causal inference regarding temporal relationships between depression and outcomes. Among 274 patients with complete data (mean age 54.0 years, 52.1% triple-negative), depression was prevalent (52.6% any depression; 17.9% CSD). Treatment uptake was 64.6% overall. Contrary to findings from high-income settings, CSD showed no significant association with treatment receipt (adjusted OR 0.95, 95% CI 0.36–2.52, p = 0.92) or mortality (adjusted HR 1.12, 95% CI 0.48–2.61, p = 0.80). Clinical factors dominated outcome prediction: lung metastasis (OR 3.63), ulceration (OR 20.39), and lymph node status significantly predicted treatment receipt, while multimodal therapy was strongly protective against mortality (HR 0.25, 95% CI 0.10–0.65). Findings should be interpreted in the context of 12.5% missing depression data and potential selection and misclassification biases. In this Nigerian cohort, CSD was common but not associated with treatment uptake or survival. These findings suggest that in resource-constrained settings where clinical triage predominates, tumor biology and treatment access may supersede psychosocial factors as outcome determinants. The substantial depression prevalence nonetheless warrants integrated mental health services within cancer care, with implications for comprehensive oncology models in LMICs.

## Introduction

Breast cancer has emerged as the most frequently diagnosed malignancy globally, with an estimated 2.3 million new cases in 2022, surpassing lung cancer for the first time [1]. This epidemiological shift carries profound implications for healthcare systems worldwide, particularly in low- and middle-income countries (LMICs) where cancer control infrastructure remains nascent and treatment access is limited [2]. The survival disparity between high-income countries (HICs) and LMICs is stark: five-year survival rates exceed 90% in North America and Western Europe but fall to approximately 40% across much of sub-Saharan Africa [3]. These disparities reflect multifactorial determinants including late-stage presentation, limited diagnostic capabilities, inadequate treatment facilities, and socioeconomic barriers to care.

Beyond healthcare system factors, accumulating evidence from HICs suggests that comorbid mental health conditions, particularly depression, may independently influence cancer treatment patterns and survival outcomes [4,5]. A comprehensive systematic review and meta-analysis by Elliott and colleagues demonstrated that breast cancer patients with pre-existing mental illness were significantly less likely to receive guideline-concordant treatment, including surgery, chemotherapy, and radiotherapy [4]. Similarly, population-based cohort studies from Scandinavian countries have documented excess mortality among cancer patients with severe mental illness, with hazard ratios ranging from 1.3 to 2.1 even after adjustment for treatment receipt [5].

**Funding:** The authors received no specific funding for this work.

**Competing interests:** The authors have declared that no competing interests exist.

Theoretical frameworks provide important context for understanding the depression–cancer outcome nexus. Beck's cognitive model of depression [6] posits that depressed individuals develop negative cognitive schemas affecting self-perception, future expectations, and interpretation of experiences, which in cancer patients may manifest as hopelessness regarding treatment efficacy, negative appraisals of prognosis, and reduced motivation for treatment-seeking. The biopsychosocial model [7] further integrates biological, psychological, and social determinants of health, suggesting that depression may influence cancer outcomes through interconnected pathways spanning neuroendocrine dysregulation, maladaptive coping behaviors, and impaired social support utilization. Lazarus and Folkman's stress-coping framework [8] additionally suggests that depression may diminish adaptive coping capacity, impairing patients' ability to navigate the complex demands of cancer treatment in resource-constrained environments.

Several mechanistic pathways have been proposed to explain the depression–cancer outcome association. From a behavioral perspective, depression may impair treatment-seeking behavior, reduce adherence to therapeutic regimens, and compromise health-promoting behaviors [9]. Biologically, depression is associated with dysregulation of the hypothalamic-pituitary-adrenal axis, chronic inflammation, and impaired immune surveillance mechanisms that may facilitate tumor progression [10]. Healthcare system factors may also contribute, as providers may consciously or unconsciously modify treatment recommendations based on perceived psychosocial complexity or anticipated adherence challenges [11].

Emerging evidence from sub-Saharan Africa has begun to illuminate the psycho-oncology landscape in the region, though significant gaps persist. Calys-Tagoe and colleagues [12] documented depression prevalence of 84.2% among Ghanaian breast cancer patients, while Ohaeri and colleagues [13] reported substantial psychosocial morbidity among Nigerian cancer patients at tertiary centers. Mohammed et al. [14] highlighted the pervasive burden of mental health comorbidities across African oncology populations, emphasizing the intersection of limited mental health infrastructure and high cancer burden. However, these studies predominantly examined depression prevalence rather than its association with treatment outcomes, leaving a critical evidence gap regarding whether the depression–outcome relationships established in HICs are generalizable to LMIC contexts.

However, the generalizability of these associations to LMIC settings remains empirically untested. The depression–cancer outcome relationship may differ fundamentally in resource-constrained healthcare environments for several reasons. First, in settings with limited treatment capacity, clinical factors related to tumor biology and resectability may dominate treatment allocation decisions, potentially minimizing the influence of psychosocial factors. Second, mental health stigma, particularly prevalent in sub-Saharan Africa, may lead to significant underreporting of depressive symptoms, affecting both diagnosis and measured associations [15]. Third, the competing mortality risks from infectious diseases and late-stage presentation may overshadow depression-related mortality effects detectable in populations with higher baseline survival.

Nigeria, Africa's most populous nation with over 220 million inhabitants, exemplifies these challenges. The country has fewer than ten functional radiotherapy centers serving the entire population, and oncology services are concentrated in urban tertiary centers [16]. Mental healthcare resources are similarly constrained, with fewer than 250 psychiatrists nationally and substantial stigma barriers to care-seeking [15]. Understanding the interplay between depression and cancer outcomes in such settings is essential for developing evidence-based, contextually appropriate comprehensive care strategies.

This study examines the association between clinically significant depression and two critical outcomes, treatment uptake and survival, among metastatic breast cancer patients at a major cancer center in Lagos, Nigeria. Metastatic breast cancer patients were specifically selected because they represent the population with the highest depression burden, the most complex treatment decisions, and the starkest outcome disparities between HIC and LMIC settings, making them the optimal population in which to test the depression–outcome relationship [17]. We hypothesized that depression would be associated with reduced treatment uptake and poorer survival, consistent with HIC findings, while recognizing that resource constraints and healthcare system factors might attenuate these relationships. Our findings contribute to the limited evidence base on psycho-oncology in sub-Saharan Africa and inform strategies for integrated cancer care in LMICs.

## Methods

### Study design and setting

This cross-sectional analysis utilized data from a prospective cohort study examining psychological distress screening tools among metastatic breast cancer patients in southwestern Nigeria. The parent study was conducted at NSIA-LUTH Cancer Centre, a specialized oncology unit within Lagos University Teaching Hospital, one of Nigeria's premier tertiary healthcare institutions. The center provides comprehensive cancer services including surgery, chemotherapy, radiotherapy, and supportive care to patients from Lagos and surrounding states. Data collection occurred between September 2020 and February 2022, spanning the COVID-19 pandemic period.

### Study population

Consecutive patients presenting with metastatic breast cancer were enrolled if they met the following criteria: (i) histopathologically confirmed breast carcinoma with clinical or radiological evidence of distant metastases; (ii) age ≥ 18 years; and (iii) ability to complete study instruments in English or with translation assistance in Yoruba, Igbo, or Hausa languages. Patients with severe cognitive impairment precluding informed consent or questionnaire completion were excluded. The study enrolled 313 patients, of whom 274 (87.5%) had complete depression assessment data and comprised the analytical sample.

### Depression assessment

Depression was assessed using the Beck Depression Inventory-II (BDI-II), a 21-item self-report instrument measuring depression severity over the preceding two weeks [18]. Each item is rated on a 4-point scale (0–3), yielding total scores ranging from 0 to 63. The BDI-II has demonstrated robust psychometric properties in oncology populations, including validation studies in sub-Saharan African cancer cohorts with Cronbach's alpha values exceeding 0.85 [19]. Standard severity categories were applied: minimal depression (0–13), mild depression (14–19), moderate depression (20–28), and severe depression (29–63) [20]. Clinically significant depression (CSD), our primary exposure, was defined as BDI-II ≥ 20, corresponding to moderate-to-severe depression warranting clinical intervention [21]. This threshold is consistent with oncology-specific validation studies, including Warmenhoven et al. [22], who validated the BDI-II ≥ 20 cutoff in advanced cancer patients, and aligns with NCCN-recommended screening thresholds for depression in oncology populations [23]. Sensitivity analyses using continuous BDI-II scores and alternative cutoffs (Distress Thermometer ≥4) provided convergent evidence supporting the robustness of this operationalization.

Depression was assessed at initial cancer center presentation, representing a cross-sectional snapshot. Therefore, temporal ordering between depression onset and treatment initiation cannot be established; CSD may represent pre-existing depression, reactive depression to the cancer diagnosis, or both, and this distinction carries important implications for interpretation. The BDI-II was administered in English, with translation assistance available in Yoruba, Igbo, and Hausa for patients with limited English proficiency. While the BDI-II has demonstrated adequate psychometric properties in sub-Saharan African cancer populations [19,22], formal linguistic validation for Nigerian languages was not performed for this study. The Distress Thermometer, as a single-item visual analog scale, is inherently less susceptible to linguistic translation challenges but may be subject to cultural interpretation differences. These considerations are discussed further in the limitations.

## Outcome measures

**Treatment uptake:** A composite binary variable indicating receipt of any cancer-directed therapy was created, including systemic chemotherapy (anthracycline-based, taxane-based, or combination regimens), targeted therapy (trastuzumab for HER2-positive disease), hormonal therapy, and surgical intervention (mastectomy or breast-conserving surgery). This composite approach acknowledged the complexity of treatment decision-making in resource-limited settings where optimal multimodal therapy may not be universally accessible.

**Survival status:** Vital status was ascertained through medical record review and direct patient/family contact at follow-up. Time-to-event was calculated from the date of cancer diagnosis to death from any cause or last known follow-up for censored observations.

## Covariates

Comprehensive clinical data were extracted from medical records and study questionnaires. Demographic variables included age and sex. Tumor characteristics encompassed histological subtype and immunohistochemical profile (estrogen receptor, progesterone receptor, HER2 status). Metastatic burden was characterized by sites of distant spread (lung, spine, liver, pleura, brain). Clinical examination findings included presence of skin ulceration, peau d'orange, and axillary lymph node status (no palpable nodes, mobile nodes, or matted nodes). Psychological distress was additionally assessed using the National Comprehensive Cancer Network Distress Thermometer (DT), a validated single-item visual analog scale (0–10) with established cutoffs for clinically significant distress (≥4) [23].

## Statistical analysis

Analyses were conducted using SAS version 9.4 (SAS Institute, Cary, NC). Descriptive statistics characterized the study population. For the primary analysis of treatment uptake, multivariable logistic regression was employed with CSD as the primary exposure variable. Potential confounders were identified through univariable analysis using a liberal $p < 0.20$ threshold and clinical judgment. To avoid overfitting, the number of covariates was limited to approximately one per 10 outcome events [24]. Quality of life demonstrated significant interaction with CSD ($p = 0.015$) and was therefore excluded from multivariable models. This decision was based on the rationale that quality of life likely lies on the causal pathway between depression and outcomes (i.e., it functions as a mediator rather than a confounder), and its inclusion would attenuate the very association under investigation, constituting inappropriate adjustment per epidemiologic guidance on distinguishing confounders from mediators [25].

Survival analysis employed Kaplan-Meier methods with log-rank testing for unadjusted comparisons and Cox proportional hazards regression for adjusted analyses. The proportional hazards assumption was verified using Schoenfeld residuals and log-log plots. Statistical significance was defined as two-sided $p < 0.05$. Post-hoc power calculations are presented for interpretive context rather than prospective justification, acknowledging the well-documented limitations of retrospective power analyses [26]. These indicated 80% power to detect odds ratios ≥2.0 for the treatment uptake

analysis given the observed CSD prevalence and event rates. Notably, the observed effect sizes were small (OR 0.95, HR 1.12) with confidence intervals spanning both protective and harmful effects, indicating genuine uncertainty rather than a definitive null finding.

The 39 patients (12.5%) with incomplete BDI-II data were compared with complete cases on all available baseline characteristics. These patients did not differ significantly in age (53.8 vs. 54.1 years, p = 0.89), treatment receipt (60.0% vs. 64.6%, p = 0.58), or mortality (11.4% vs. 14.6%, p = 0.61), suggesting that missingness was likely random rather than systematic, though non-random missingness cannot be definitively excluded [27,28].

## Ethical considerations

The study was approved by the Lagos University Teaching Hospital Health Research Ethics Committee (LUTH-HREC, ADM/DCST/HREC/APP/7058). All participants provided written informed consent. Data were de-identified prior to analysis, and confidentiality was maintained throughout in accordance with the Declaration of Helsinki. This study represents a secondary analysis of de-identified patient data from the original study. All identifying information was removed prior to data sharing, and researchers had access only to anonymized data. Data were first accessed for this secondary analysis on October 27, 2024.

## Results

### Study population characteristics

Of 313 enrolled patients, 274 (87.5%) had complete BDI-II data and comprised the analytical cohort. The study population was predominantly female (99.4%) with a mean age of 54.0 years (SD 13.0, median 53.0). The near-exclusive female composition reflects the disease epidemiology of breast cancer but limits generalizability to male breast cancer patients, who may experience distinct psychosocial and treatment-seeking patterns. Invasive ductal carcinoma was the dominant histology (93.6%), and triple-negative breast cancer was the most common immunophenotype (52.1%), followed by hormone receptor-positive/HER2-negative disease (21.9%). Lung metastases were most prevalent (53.2%), followed by spine (48.4%), pleura (18.3%), liver (13.5%), and brain (12.8%). Physical examination revealed ulceration in 8.3%, peau d'orange in 9.9%, and significant axillary lymphadenopathy in 53.2% (mobile nodes: 29.2%; matted nodes: 24.0%). Complete demographic and clinical characteristics are presented in **Table 1**.

### Depression and psychological distress burden

Depression was highly prevalent in this metastatic breast cancer population. Overall, 144 patients (52.6%) exhibited depressive symptoms of any severity, and 49 (17.9%) met criteria for CSD (BDI-II ≥ 20). The severity distribution was: minimal/no depression (BDI-II < 14), 130 (47.4%); mild depression (BDI-II 14–19), 95 (34.7%); moderate depression (BDI-II 20–28), 45 (16.4%); and severe depression (BDI-II 29–63), 4 (1.4%). The predominance of mild-to-moderate depressive symptoms (34.7% mild, 16.4% moderate) with relatively few cases of severe depression (1.4%) suggests a preponderance of adjustment-related depressive responses rather than severe depressive disorder, consistent with psychological adaptation models in cancer populations [29]. Mean BDI-II scores differed substantially between groups: 8.2 ± 4.1 in the non-CSD group versus 23.1 ± 3.8 in the CSD group (p < 0.001). Concordantly, significant psychological distress (DT ≥ 4) was present in 46.0% of participants, with a trend toward higher distress scores in the CSD group that did not reach statistical significance (57.1% vs. 43.6%, p = 0.10). Depression characteristics stratified by CSD status are presented in **Table 2**.

### Association between depression and treatment uptake

Overall, 177 patients (64.6%) received cancer-directed treatment. Treatment modalities included surgery (n = 161, 51.6%), anthracycline-based chemotherapy (AC: n = 45; CAF: n = 28; FEC: n = 19), cyclophosphamide-based regimens

**Table 1. Patient demographics and clinical characteristics (N = 313).**

| Characteristic | N (%) or Mean±SD |
|---|---|
| **Demographics** | |
| Age, years | 54.0 ± 13.0 |
| Female sex | 311 (99.4%) |
| **Tumor Characteristics** | |
| Histology | |
| Invasive ductal carcinoma | 291 (93.6%) |
| Invasive lobular carcinoma | 7 (2.3%) |
| Other | 13 (4.2%) |
| Immunohistochemistry | |
| ER+/PR+/HER2- | 68 (21.9%) |
| ER-/PR-/HER2- (Triple negative) | 162 (52.1%) |
| ER+/PR+/HER2+ (Triple positive) | 10 (3.2%) |
| ER-/PR-/HER2+ | 21 (6.7%) |
| Other combinations | 50 (16.1%) |
| **Metastatic Sites** | |
| Lung | 166 (53.2%) |
| Spine | 151 (48.4%) |
| Liver | 42 (13.5%) |
| Pleura | 57 (18.3%) |
| Brain | 40 (12.8%) |
| **Clinical Findings** | |
| Ulceration | 26 (8.3%) |
| Peau d'orange | 31 (9.9%) |
| Mobile axillary lymph nodes | 91 (29.2%) |
| Matted axillary lymph nodes | 75 (24.0%) |
| **Treatment Received** | |
| Any treatment | 187 (59.9%) |
| Surgery only | 34 (10.9%) |
| Chemotherapy only | 31 (9.9%) |
| Combined therapy | 122 (39.1%) |

(CMF: n = 10), taxane-based therapy (n = 27), and targeted therapy with trastuzumab (n = 12). Many patients received multimodal therapy. Strikingly, treatment receipt was nearly identical between depression groups: 65.3% (32/49) of CSD patients and 64.4% (145/225) of non-CSD patients received treatment (unadjusted OR 1.04, 95% CI 0.52–2.07, p = 0.92).

In multivariable logistic regression adjusting for age, distress, and clinical factors, CSD remained non-significantly associated with treatment receipt (adjusted OR 0.95, 95% CI 0.36–2.52, p = 0.92). Depression assessment and treatment status were ascertained contemporaneously at the cross-sectional evaluation; thus, the temporal sequence between depression onset and treatment initiation cannot be determined. In contrast, clinical indicators of disease extent and resectability demonstrated strong associations with treatment receipt. Patients with lung metastases were significantly more likely to receive treatment (OR 3.63, 95% CI 1.59–8.29, p = 0.002), as were those with ulcerating tumors (OR 20.39, 95% CI 2.39–173.92, p = 0.006). The very wide confidence interval for ulceration reflects sparse data for this predictor (n = 26, 8.3%) and should be interpreted as indicating the direction and approximate magnitude of association rather than a precise point estimate, consistent with known small-sample bias in logistic regression [30]. Axillary lymph node status

**Global Public Health**

PLOS

**Table 2. Depression characteristics and clinical outcomes by CSD status (N = 274).**

| Characteristic | No CSD (N = 225) | CSD (N = 49) | p-value |
|---|---|---|---|
| **Depression Severity** | | | |
| BDI-II Score, mean±SD | 8.2±4.1 | 23.1±3.8 | <0.001 |
| Minimal/No (0–13) | 130 (57.8%) | 0 (0%) | — |
| Mild (14–19) | 95 (42.2%) | 0 (0%) | — |
| Moderate (20–28) | 0 (0%) | 45 (91.8%) | — |
| Severe (29–63) | 0 (0%) | 4 (8.2%) | — |
| **Clinical Outcomes** | | | |
| Treatment received | 145 (64.4%) | 32 (65.3%) | 0.92 |
| Mortality | 33 (14.7%) | 7 (14.3%) | 0.95 |
| Distress score ≥4 | 98 (43.6%) | 28 (57.1%) | 0.10 |
| **Follow-up** | | | |
| Median follow-up, days | 756 | 742 | 0.68 |
| Range, days | 45-1095 | 52-1003 | |

**Abbreviations:** CSD, clinically significant depression; BDI-II, Beck Depression Inventory-II.

**Note:** BDI-II severity categories are deterministically defined by the CSD threshold (≥20); individual category p-values are therefore redundant and have been removed. The reported p-value for BDI-II score represents the continuous score comparison between groups. "—" indicates p-value not applicable.

was the strongest predictor: mobile nodes (OR 20.33, 95% CI 7.18–57.58, p<0.001) and matted nodes (OR 7.84, 95% CI 3.05–20.15, p<0.001) were both strongly associated with treatment receipt compared to patients without palpable lymphadenopathy. The model demonstrated good discrimination (c-statistic = 0.84). Full results are presented in **Table 3**.

## Depression and survival outcomes

During follow-up (median 756 days, IQR 45–1095), 40 patients (14.6%) died. Mortality was similarly distributed between depression groups: 7/49 (14.3%) in the CSD group and 33/225 (14.7%) in the non-CSD group. Kaplan-Meier survival analysis revealed virtually superimposable survival curves between CSD and non-CSD patients throughout the observation period (log-rank p = 0.95; **Fig 1**). Survival probabilities at 200, 400, 600, and 800 days were nearly identical between groups (**S1 Table**).

**Table 3. Factors associated with treatment receipt: Multivariable logistic regression.**

| Variable | Adjusted OR | 95% CI | p-value |
|---|---|---|---|
| **Depression and Distress** | | | |
| Clinically significant depression | 0.949 | 0.357-2.523 | 0.917 |
| NCCN distress score | 0.899 | 0.710-1.138 | 0.377 |
| **Clinical Factors** | | | |
| Age (per year) | 1.004 | 0.977-1.030 | 0.791 |
| Lung metastasis | 3.626 | 1.586-8.290 | 0.002 |
| Pleura metastasis | 0.395 | 0.154-1.014 | 0.053 |
| Ulceration present | 20.389 | 2.390-173.917 | 0.006 |
| Mobile axillary lymph nodes | 20.330 | 7.179-57.575 | <0.001 |
| Matted axillary lymph nodes | 7.843 | 3.053-20.146 | <0.001 |

Model c-statistic = 0.84, indicating good discrimination.

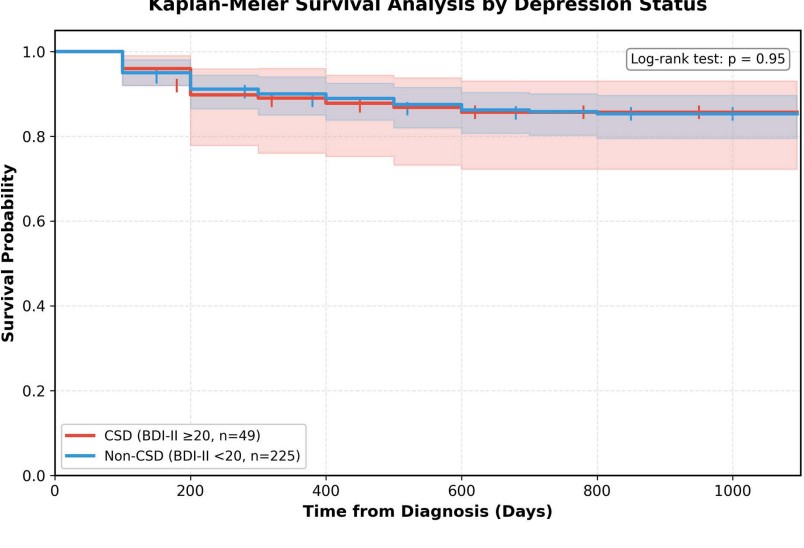

**Fig 1. Kaplan-Meier survival analysis by depression status.** Survival curves comparing metastatic breast cancer patients with clinically significant depression (CSD; BDI-II ≥ 20, red line, n = 49) versus those without CSD (blue line, n = 225) at NSIA-LUTH Cancer Centre, Lagos, Nigeria. Time zero represents date of cancer diagnosis. Shaded areas indicate 95% confidence intervals. Vertical tick marks denote censored observations. The number-at-risk table shows patients remaining under observation at each time point. No significant difference in survival was observed between groups (log-rank p = 0.95). Median follow-up was 756 days with 40 total deaths (14.6%). CSD, clinically significant depression; BDI-II, Beck Depression Inventory-II.

Cox proportional hazards analysis confirmed the absence of association between CSD and mortality. The unadjusted hazard ratio for CSD versus non-CSD was 0.97 (95% CI 0.43–2.19, p = 0.95), and the adjusted hazard ratio was 1.12 (95% CI 0.48–2.61, p = 0.80) after controlling for age, treatment, distress, and clinical covariates. With only 40 deaths (7 in the CSD group), the analysis had limited statistical power to detect modest hazard ratios, and the wide confidence interval (0.48–2.61) spans both protective and harmful effects, indicating that a clinically meaningful association cannot be excluded. Clinical factors demonstrated stronger associations: ulceration (HR 19.26, 95% CI 2.48–149.85, p = 0.005) and lymph node involvement predicted increased mortality, while receipt of treatment, particularly combined chemotherapy and surgery, was strongly protective (HR 0.25, 95% CI 0.10–0.65, p = 0.004). Lung metastasis was associated with a 91.4% lower hazard of mortality (adjusted HR 0.086, 95% CI 0.019–0.397, p = 0.002). This counterintuitive finding may reflect multiple mechanisms: (i) ascertainment bias, wherein patients with lung metastases may have been identified through routine imaging, indicating earlier detection within the metastatic cascade; (ii) differential access, as pulmonary symptoms may prompt earlier presentation to oncology services; and (iii) immortal time bias, as patients surviving long enough for lung metastasis documentation may represent inherently better-prognosis individuals. This finding warrants cautious interpretation and prospective validation. Complete Cox regression results are presented in **Table 4**.

### Sensitivity analyses

Several sensitivity analyses were conducted to test the robustness of findings. When depression was modeled as a continuous variable, BDI-II scores showed no significant association with treatment receipt (OR per 5-point increase: 1.02, 95% CI 0.89–1.17, p = 0.78) or mortality (HR per 5-point increase: 0.96, 95% CI 0.79–1.16, p = 0.66). Analysis using the Distress Thermometer as an alternative psychological measure yielded consistent results: high distress (DT ≥ 4) was

**Table 4. Cox proportional hazards analysis for mortality.**

| Variable | Unadjusted HR (95% CI) | p-value | Adjusted HR (95% CI) | p-value |
|---|---|---|---|---|
| **Depression Status** | | | | |
| CSD vs Non-CSD | 0.97 (0.43-2.19) | 0.95 | 1.12 (0.48-2.61) | 0.80 |
| **Treatment Factors** | | | | |
| Any treatment vs none | 0.45 (0.23-0.88) | 0.02 | 0.41 (0.20-0.84) | 0.01 |
| Combined therapy vs none | 0.28 (0.12-0.68) | 0.005 | 0.25 (0.10-0.65) | 0.004 |
| Surgery only vs none | 0.72 (0.31-1.68) | 0.45 | 0.68 (0.28-1.65) | 0.39 |
| Chemotherapy only vs none | 0.58 (0.21-1.62) | 0.30 | 0.52 (0.18-1.48) | 0.22 |
| **Patient Factors** | | | | |
| Age (per 10 years) | 1.18 (0.89-1.56) | 0.25 | 1.22 (0.91-1.64) | 0.18 |
| Distress score ≥4 | 1.35 (0.70-2.60) | 0.37 | 1.28 (0.65-2.52) | 0.47 |
| **Clinical Factors** | | | | |
| Lung metastasis† | 0.092 (0.021-0.405) | 0.003 | 0.086 (0.019-0.397) | 0.002 |
| Ulceration present | 17.850 (2.285-139.452) | 0.006 | 19.262 (2.476-149.855) | 0.005 |
| Mobile axillary lymph nodes | 18.245 (2.421-137.491) | 0.005 | 19.221 (2.550-144.875) | 0.004 |
| Matted axillary lymph nodes | 13.821 (1.022-186.985) | 0.048 | 14.540 (1.068-197.928) | 0.045 |

Adjusted model includes age, treatment status, distress score, and available clinical covariates.

**Abbreviations:** HR, hazard ratio; CI, confidence interval; CSD, clinically significant depression.

† Lung metastasis HR of 0.086 corresponds to a 91.4% lower hazard of mortality compared with the reference group (no lung metastasis). This counterintuitive finding may reflect ascertainment bias (earlier detection through routine imaging), differential access (pulmonary symptoms prompting earlier oncology presentation), or immortal time bias (patients surviving long enough for documentation representing inherently better-prognosis individuals). See Discussion for detailed interpretation.

not significantly associated with treatment uptake (64.3% vs. 64.9%, p = 0.93) or survival (HR 1.28, 95% CI 0.65–2.52, p = 0.47). Missing data analysis revealed that the 35 patients with incomplete BDI-II data had similar baseline characteristics to complete cases, including age (53.8 vs. 54.1 years, p = 0.89), treatment receipt (60.0% vs. 64.6%, p = 0.58), and mortality (11.4% vs. 14.6%, p = 0.61), suggesting minimal selection bias.

## Discussion

This cross-sectional analysis of metastatic breast cancer patients at a major Nigerian cancer center yielded three principal findings. First, depression was highly prevalent, with over half of patients exhibiting depressive symptoms and nearly one-fifth meeting criteria for clinically significant depression warranting intervention. Second, contrary to findings from high-income settings, CSD was not associated with treatment uptake, patients with and without significant depression received cancer-directed therapy at virtually identical rates. Third, depression showed no association with survival outcomes over the follow-up period, with clinical factors and treatment access emerging as dominant outcome predictors. These findings have important implications for understanding psycho-oncology within LMIC contexts and for developing contextually appropriate comprehensive cancer care models.

### Depression burden in context

The 17.9% prevalence of CSD and 52.6% prevalence of any depression aligns with global estimates from meta-analyses reporting depression rates of 15–25% among breast cancer patients [31,32]. These rates are also broadly consistent with emerging data from the African continent, including Calys-Tagoe et al.'s [12] report of 84.2% depression prevalence among Ghanaian breast cancer patients and Ohaeri et al.'s [13] documentation of substantial psychosocial morbidity in

Nigerian oncology populations. However, several factors suggest these estimates may underestimate true depression burden in this population. Mental health stigma remains substantial in Nigeria, with studies documenting that over half of healthcare providers hold stigmatizing attitudes toward mental illness [15]. Patients may consequently underreport symptoms on self-administered instruments. Additionally, depression in cancer patients often manifests with somatic symptoms, fatigue, appetite disturbance, sleep disruption, that overlap with cancer symptoms and treatment effects, potentially leading to misattribution and underrecognition [22]. The predominance of triple-negative disease (52.1%) in our cohort, higher than typically observed in Western populations, may reflect both biological differences in African breast cancer epidemiology and the aggressive disease biology associated with delayed presentation.

It is important to note that the BDI-II identifies symptom severity levels exceeding validated thresholds rather than formal psychiatric diagnoses per DSM-5 or ICD-11 criteria. The absence of structured clinical interviews (e.g., SCID) means that our CSD classification captures self-reported symptom burden, which may differ from clinical depression diagnosed by mental health professionals. This distinction is relevant because self-report instruments may both over-identify (through somatic symptom overlap with cancer) and under-identify (through stigma-related underreporting) clinically meaningful depression in this population.

### The null depression–treatment association: A resource-constraint hypothesis

Our finding that CSD was not associated with treatment uptake contrasts sharply with evidence from HICs, where systematic reviews document significantly reduced treatment receipt among cancer patients with mental illness [4]. We propose that this divergence may reflect differences in treatment allocation mechanisms between resource-abundant and resource-constrained settings. In HICs, where treatment capacity generally exceeds demand, psychosocial factors may influence patient preferences, provider recommendations, and treatment adherence in ways that translate to measurable disparities. In contrast, in settings like Nigeria where treatment capacity is severely limited, fewer than ten radiotherapy centers serve over 220 million people [16], clinical triage based on tumor biology, disease stage, and anticipated treatment response may predominate, potentially overriding psychosocial considerations.

The strong associations observed between clinical factors and treatment receipt support this hypothesis. The very large odds ratios for lymph node status (OR 20.3 for mobile nodes) and ulceration (OR 20.4) likely reflect clinical decision algorithms that heavily weight tumor resectability and disease extent. The conceptual framework illustrated in **Fig 2** depicts these hypothesized relationships. In effect, the clinical imperative to allocate scarce treatment resources optimally may create conditions that are paradoxically more equitable access across psychosocial strata, though at the cost of overall treatment availability. This resource-constraint hypothesis represents a tentative explanatory framework requiring prospective validation with health system data, including provider decision-making processes and treatment allocation criteria, rather than an established mechanism.

### Survival patterns and competing determinants

The absence of survival differences between CSD and non-CSD patients (14.3% vs. 14.7% mortality) contrasts with population-based studies from HICs demonstrating significant excess mortality among cancer patients with severe mental illness [5,33]. Several factors may explain this finding. First, the cross-sectional assessment of depression at cancer presentation captures a mixture of reactive depression (temporally proximate to cancer diagnosis) and pre-existing depression. Evidence suggests that chronic or pre-existing depression has stronger associations with cancer outcomes than acute reactive depression [4], but our design could not distinguish these constructs. Second, in a population with high baseline mortality from metastatic disease, the marginal effect of depression may be difficult to detect, particularly with limited statistical power. Third, competing risks from the aggressive tumor biology characteristic of this population, predominantly triple-negative and late-stage disease, may overwhelm any depression-attributable mortality increment.

## Conceptual Framework: Depression–Cancer Outcome Pathways in LMIC Settings

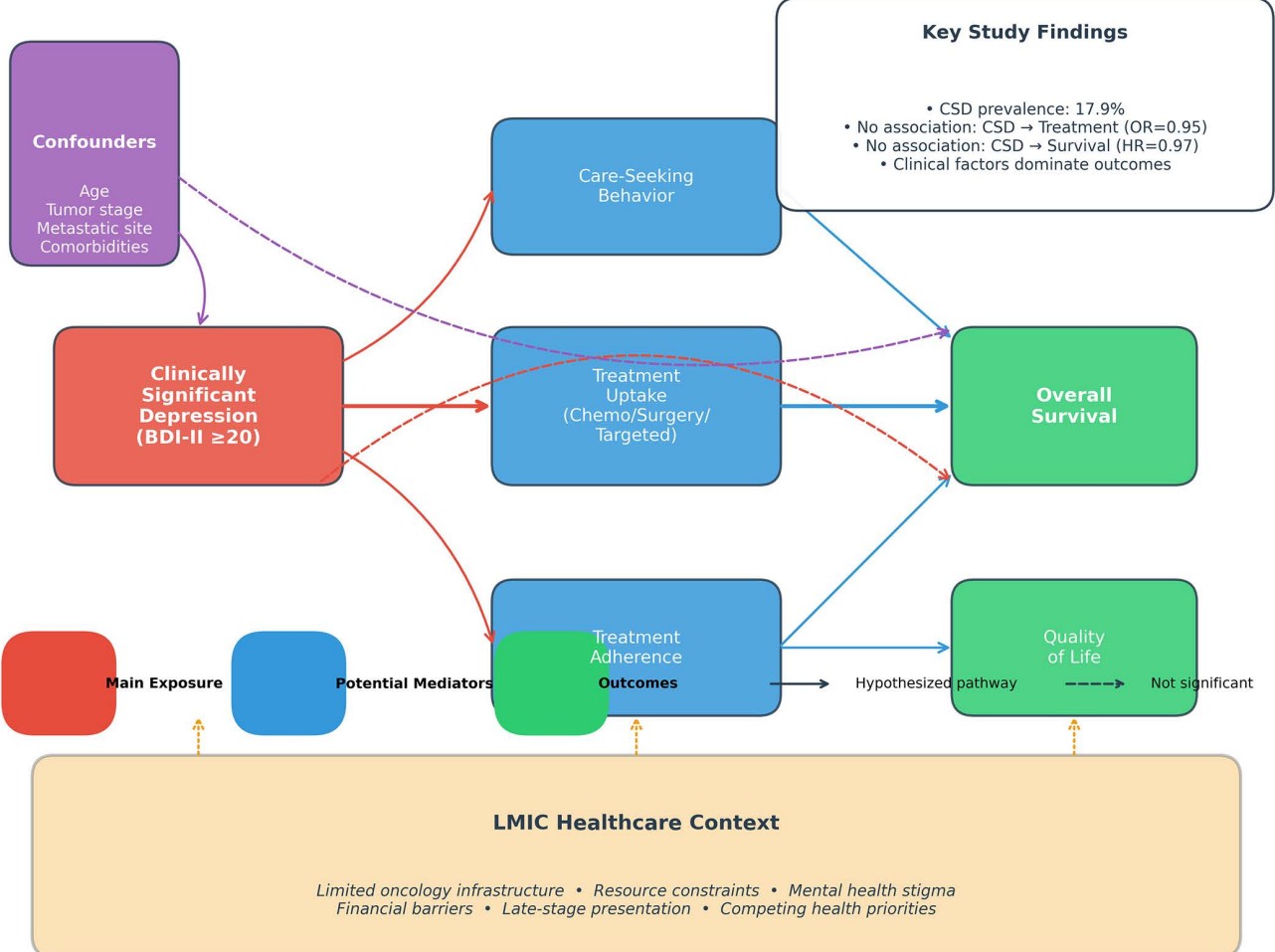

**Fig 2. Conceptual framework: Depression–cancer outcome pathways in LMIC settings.** This framework illustrates the hypothesized pathways linking clinically significant depression to cancer outcomes in resource-limited healthcare contexts. The main exposure (depression, red) may influence outcomes through multiple potential mediators (blue) including treatment uptake, care-seeking behavior, and treatment adherence. In high-income settings, these pathways typically demonstrate significant associations with survival outcomes (green). However, in LMIC contexts (orange shading), structural factors including limited oncology infrastructure, resource constraints, mental health stigma, and financial barriers may attenuate these relationships. Our study found no significant association between CSD and treatment uptake (OR=0.95) or survival (HR=0.97), suggesting that clinical factors dominate outcome determination in resource-constrained settings where treatment allocation is driven primarily by tumor characteristics and disease severity. The dashed direct effect pathway was not significant in our analysis. Confounders (purple) including age, tumor stage, metastatic sites, and comorbidities were adjusted for in all analyses.

Notably, treatment access emerged as a dominant survival determinant. The 75% mortality reduction associated with combined chemotherapy and surgery (HR 0.25) underscores the critical importance of multimodal treatment access, rather than psychosocial optimization, as the proximate driver of survival improvement in this setting. This finding aligns with the broader literature on cancer care disparities in LMICs [3] and suggests that interventions targeting treatment access may yield greater survival gains than psychosocial interventions alone.

## Alternative explanations for null findings

Several alternative explanations for the null depression–outcome associations warrant consideration beyond the resource-constraint hypothesis. First, measurement bias from potential underreporting due to stigma and cultural response patterns may have attenuated true associations. Second, misclassification bias is a concern; if patients with genuine CSD scored below the BDI-II ≥ 20 threshold due to symptom underreporting, they would be classified into the non-CSD group, diluting the exposure contrast and biasing results toward the null. Third, residual confounding from unmeasured variables, including socioeconomic status, insurance coverage, social support networks, health literacy, and traditional medicine use, cannot be excluded. Fourth, the BDI-II cutoff of ≥20, while validated in Western oncology populations, may not optimally capture the construct of clinically significant depression in this Nigerian cohort, where somatic symptom expression and cultural manifestations of distress may differ from Western presentations [34,35]. These alternative explanations collectively suggest caution in interpreting the null findings as definitive evidence of absence.

## Strengths and limitations

This study has several strengths. It represents one of the largest investigations of depression and cancer outcomes from sub-Saharan Africa, utilizing validated instruments with established psychometric properties in African populations. The comprehensive clinical data enabled robust adjustment for clinicopathological confounders. The single-center design, while limiting generalizability, ensured consistency in clinical practice patterns and data collection.

Important limitations warrant consideration. The cross-sectional design precludes causal inference regarding the temporal relationship between depression and outcomes. Statistical power was limited for detecting modest effect sizes, and larger multicenter studies are needed to exclude clinically meaningful associations. The relatively short median follow-up (756 days) may have been insufficient to detect survival differences that emerge over longer horizons [33]. Cultural factors may affect symptom expression and reporting on Western-developed instruments, despite validation efforts. Formal linguistic validation of the BDI-II for Nigerian languages (Yoruba, Igbo, Hausa) was not performed, and cross-cultural equivalence of the instrument cannot be assumed. Finally, unmeasured confounding, including socioeconomic status, social support, and health literacy, cannot be excluded.

A particularly important limitation relates to potential misclassification bias arising from mental health stigma and underreporting. In the Nigerian sociocultural context, where mental health stigma remains substantial [15], patients may underreport depressive symptoms on self-administered instruments. This would result in differential misclassification, whereby truly depressed patients scoring below the BDI-II ≥ 20 threshold would be classified into the non-CSD group, effectively diluting the CSD group and attenuating any true association between depression and outcomes toward the null. This non-differential misclassification of the exposure represents a bias toward the null hypothesis and may partly explain the absence of observed associations. The potential magnitude of this bias is difficult to quantify without a gold-standard clinical interview comparison, but the documented prevalence of mental health stigma in Nigerian healthcare settings suggests it may be non-trivial [34].

## Clinical and policy implications

Despite the null findings for depression-outcome associations, the substantial depression prevalence observed carries important implications for cancer care delivery. First, depression affects quality of life and patient-reported outcomes regardless of effects on survival, warranting attention on humanitarian and patient-centered care grounds [36]. Second, simple screening tools such as the Distress Thermometer can feasibly be implemented in resource-constrained settings to identify patients requiring psychosocial support [23]. Third, our findings suggest that in severely resource-constrained settings, interventions to expand treatment capacity and reduce structural barriers to care may yield greater survival benefits than targeted psychosocial interventions, though these approaches need not be mutually exclusive. Importantly, our findings should not be interpreted as evidence that psychosocial interventions lack survival benefit in LMIC settings; rather,

they suggest that the relationship between depression and cancer outcomes may be context-dependent and mediated by health system characteristics. The substantial depression prevalence underscores the need for integrated mental health services on humanitarian, quality-of-life, and patient-centered grounds, regardless of measured survival associations.

Practical implementation of mental health integration within Nigerian oncology settings warrants consideration. Brief validated screening tools (DT, PHQ-2) require minimal training and can be administered in under 5 minutes, making them feasible for routine use in busy tertiary cancer centers. Task-shifting models utilizing trained oncology nurses for initial screening and psychoeducation have demonstrated effectiveness in similar LMIC contexts [37]. At NSIA-LUTH and comparable Nigerian tertiary centers, mental health screening could be integrated into existing treatment workflows at the point of initial oncology consultation, with referral pathways to psychiatry or clinical psychology for patients exceeding screening thresholds. Digital health innovations, including mobile-based psychoeducation and remote psychological support platforms, offer additional scalable approaches to addressing the substantial mental health burden in settings with limited specialized mental health workforce [38].

### Future research directions

Future studies should employ prospective longitudinal designs with serial depression assessments to capture the temporal dynamics of the depression–cancer outcome relationship. Multicenter collaborations across African cancer centers could provide adequate power to detect modest associations and improve generalizability. Research examining health-care system factors mediating these relationships, including provider decision-making processes, treatment allocation criteria, and patient navigation, would inform intervention development. Finally, implementation studies evaluating integrated mental health and cancer care models adapted for LMIC contexts are needed to establish evidence-based approaches to addressing the substantial psychosocial burden we observed.

### Conclusions

This analysis of Nigerian metastatic breast cancer patients revealed substantial depression prevalence but no significant associations between clinically significant depression and treatment uptake or survival outcomes. These findings, divergent from HIC literature, suggest that in resource-constrained settings where clinical triage predominates, tumor biology and treatment access may supersede psychosocial factors as primary outcome determinants. The 17.9% prevalence of clinically significant depression nonetheless represents a substantial unmet mental health need warranting integration of psychological screening and support services into cancer care. Future prospective multicenter studies are needed to definitively characterize depression–cancer outcome relationships across diverse African populations and to inform evidence-based models for comprehensive cancer care in LMICs.

### Supporting information

**S1 Table. Kaplan-Meier survival probabilities by depression status.**
(DOCX)

### Acknowledgments

We thank the patients and families who participated in this study, as well as the healthcare providers at NSIA-LUTH Cancer Centre who facilitated data collection.

### Author contributions

**Conceptualization:** Adewumi Alabi, Abdulrazzaq Lawal, Oluwaseun Adebayo Bamodu.

**Data curation:** Emma Wisniewski, Adewumi Alabi, Anthonia Sowunmi, Chen-Chih Chung, Sumaiya Nezam, Sylvia Shirima, Donaldson F. Conserve, Oluwaseun Adebayo Bamodu.

**Formal analysis:** Emma Wisniewski, Adewumi Alabi, Chen-Chih Chung, Sumaiya Nezam, Sylvia Shirima, Donaldson F. Conserve, Oluwaseun Adebayo Bamodu.

**Investigation:** Adewumi Alabi, Abdulrazzaq Lawal, Anthonia Sowunmi, Bolanle Adegboyega.

**Methodology:** Abdulrazzaq Lawal, Anthonia Sowunmi, Bolanle Adegboyega, Chen-Chih Chung, Oluwaseun Adebayo Bamodu.

**Project administration:** Adewumi Alabi, Donaldson F. Conserve, Oluwaseun Adebayo Bamodu.

**Resources:** Donaldson F. Conserve.

**Supervision:** Oluwaseun Adebayo Bamodu.

**Validation:** Oluwaseun Adebayo Bamodu.

**Visualization:** Oluwaseun Adebayo Bamodu.

**Writing – original draft:** Emma Wisniewski, Oluwaseun Adebayo Bamodu.

**Writing – review & editing:** Oluwaseun Adebayo Bamodu.

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
