## [Decision Letter · Decision Letter 0]

5 Mar 2026

PGPH-D-26-00350

Depression and Cancer Outcomes in Resource-Limited Settings: A Cross-Sectional Analysis of Treatment Uptake and Survival Among Metastatic Breast Cancer Patients in Lagos, Nigeria

Dear Dr Oluwaseun Adebayo Bamodu,

Thank you for submitting your manuscript to PLOS Global Public Health. After careful consideration, we feel that it has merit but does not fully meet PLOS Global Public Health’s publication criteria as it currently stands. Therefore, we invite you to submit a revised version of the manuscript that addresses the points raised during the review process.

We look forward to receiving your revised manuscript.

Kind regards,

Revathy Sudhakar, M.Sc., M.Phil.,

Guest Editor

Journal Requirements:

Additional Editor Comments (if provided):

Reviewers' comments:

Reviewer's Responses to Questions

**Comments to the Author**

1. Does this manuscript meet PLOS Global Public Health’s publication criteria? Is the manuscript technically sound, and do the data support the conclusions? The manuscript must describe methodologically and ethically rigorous research with conclusions that are appropriately drawn based on the data presented.? Is the manuscript technically sound, and do the data support the conclusions? The manuscript must describe methodologically and ethically rigorous research with conclusions that are appropriately drawn based on the data presented.

Reviewer #1: Yes

Reviewer #2: Yes

2. Has the statistical analysis been performed appropriately and rigorously?

Reviewer #1: Yes

Reviewer #2: Yes

3. Have the authors made all data underlying the findings in their manuscript fully available (please refer to the Data Availability Statement at the start of the manuscript PDF file)?

The PLOS Data policy requires authors to make all data underlying the findings described in their manuscript fully available without restriction, with rare exception. The data should be provided as part of the manuscript or its supporting information, or deposited to a public repository. For example, in addition to summary statistics, the data points behind means, medians and variance measures should be available. If there are restrictions on publicly sharing data—e.g. participant privacy or use of data from a third party—those must be specified.requires authors to make all data underlying the findings described in their manuscript fully available without restriction, with rare exception. The data should be provided as part of the manuscript or its supporting information, or deposited to a public repository. For example, in addition to summary statistics, the data points behind means, medians and variance measures should be available. If there are restrictions on publicly sharing data—e.g. participant privacy or use of data from a third party—those must be specified.

Reviewer #1: Yes

Reviewer #2: Yes

4. Is the manuscript presented in an intelligible fashion and written in standard English?

Reviewer #1: Yes

Reviewer #2: Yes

Reviewer #1: This study offers new insights into psycho-oncology in sub-Saharan Africa by examining the association between clinically significant depression (CSD) and treatment uptake and survival among patients with metastatic breast cancer in Nigeria. My observations are given below:

(1) The study title is okay.

(2) This abstract provides relevant analyses to fill a significant LMIC gap. To support the BDI-II cutoff validity in Nigeria, explain the limitations of cross-sectional design for survival inference, and talk about how selection bias and incomplete data may affect results and generalizability.

(3) The introduction is well-organized and explains the disparities in survival, the worldwide burden of breast cancer, and the importance of looking at depression in LMIC settings. The HIC evidence review and suggested mechanisms are succinct and pertinent. To strengthen the conceptual framework and make clear why metastatic patients were chosen in particular, the authors should, nevertheless, more fully incorporate psychological theory. If available, a deeper reading of the body of existing African psycho-oncology literature would improve contextual grounding and show the study's uniqueness outside of service-level limitations.

(4) Temporal ordering needs to be clarified because the cross-sectional analysis of a prospective cohort restricts the causal inference between depression and outcomes. It is necessary to justify and evaluate selection bias when handling the 12.5 percent of missing BDI-II data. It is necessary to elaborate on the cultural validation and translation processes for the BDI-II and Distress Thermometer in Nigerian languages. Oncology-specific cutoffs should be used to support the definition of CSD as ≥20. A critical reexamination of the exclusion of quality of life because of interaction is warranted. Power calculations should be interpreted carefully because they are post-hoc.

(5) Although the results are clinically detailed and well-organized, there needs to be more integration of psychological interpretation. It is important to recognize that the sample's preponderance of women restricts generalizability. Although it is appropriate to report effect sizes, wide confidence intervals (e.g., lung metastases, ulceration) point to possible instability and small-event bias. Beyond assumed selection bias, further research is necessary to fully understand the unexpected protective effect of lung metastases. Make the time sequence between the diagnosis of depression and the start of treatment more clear. Talk about the statistical power to identify correlations between depression and mortality in light of low death rates.

(6) With careful interpretation of null findings, the discussion is logical and contextually based. Although the resource-constraint theory makes sense conceptually, it is still theoretical in the absence of concrete health system data; temper causal language. Make it clear if the instruments recorded symptom thresholds or clinical diagnoses. Inferences would be strengthened by a greater consideration of alternative explanations, such as measurement bias and residual confounding. While treatment access is rightly prioritized in the policy implications, it is not implied that psychosocial interventions are not relevant for survival. Give a brief explanation of the feasibility and clinical integration pathways in Nigerian oncology settings.

(7) The references are ok. Please correct it by following the journal’s guidelines. The tables and figures should be prepared following standard guidelines.

Reviewer #2: 1. In Table 2, the p-values for BDI-II categories are redundant with the overall group difference; consider simplifying this.

2. The hazard ratio (0.086) of lung metastasis is striking. Consider presenting as "91% lower hazard of the event compared with the reference group" for clarity.

3. For the mental health stigma and underreporting, the authors acknowledged stigma, but the implications of this have not been fully explored. If patients underreport depressive symptoms, the CSD group may be misclassified, where some truly depressed patients are categorised as being in the non-CSD group, biasing the results. I suggest adding this to the manuscript's limitations section.

**Do you want your identity to be public for this peer review?** For information about this choice, including consent withdrawal, please see our Privacy Policy..

Reviewer #1: **Yes:** Gyanesh Kumar TiwariGyanesh Kumar TiwariGyanesh Kumar TiwariGyanesh Kumar Tiwari

Reviewer #2: **Yes:** Taiwo Olufemi AbionaTaiwo Olufemi AbionaTaiwo Olufemi AbionaTaiwo Olufemi Abiona

---

## [Editor Report · Decision Letter 1]

16 Mar 2026

Depression and Cancer Outcomes in Resource-Limited Settings: A Cross-Sectional Analysis of Treatment Uptake and Survival Among Metastatic Breast Cancer Patients in Lagos, Nigeria

PGPH-D-26-00350R1

Dear Oluwaseun Adebayo Bamodu,

We are pleased to inform you that your manuscript 'Depression and Cancer Outcomes in Resource-Limited Settings: A Cross-Sectional Analysis of Treatment Uptake and Survival Among Metastatic Breast Cancer Patients in Lagos, Nigeria' has been provisionally accepted for publication in PLOS Global Public Health.

Best regards,

Revathy Sudhakar, M.Sc., M.Phil.,

Guest Editor

Dear authors,

I sincerely appreciate the efforts carried out in addressing each comment systematically.

The manuscript overall reads well and will contribute significantly to the scientific community.

Best wishes,

Revathy